# Metabolic Interventions in Tumor Immunity: Focus on Dual Pathway Inhibitors

**DOI:** 10.3390/cancers15072043

**Published:** 2023-03-29

**Authors:** Min Chen, Huanrong Lan, Shiya Yao, Ketao Jin, Yun Chen

**Affiliations:** 1Department of Colorectal Surgery, Sir Run Run Shaw Hospital, Zhejiang University School of Medicine, Hangzhou 310016, China; 2Department of Surgical Oncology, Affiliated Hangzhou Cancer Hospital, Zhejiang University School of Medicine, Hangzhou 310002, China; 3Department of Colorectal Surgery, Affiliated Jinhua Hospital, Zhejiang University School of Medicine, Jinhua 321000, China; 4Department of Colorectal Surgery, Xinchang People’s Hospital, Affiliated Xinchang Hospital, Wenzhou Medical University, Xinchang 312500, China

**Keywords:** metabolic intervention, dual inhibitor, metabolic reprogramming, cancer therapy

## Abstract

**Simple Summary:**

Metabolic reprogramming is one of tumor and immune cells’ most significant metabolic alterations. Moreover, metabolic-related signaling pathways, such as phosphoinositide 3-kinases (PI3Ks), the mammalian target of rapamycin (mTOR), can induce growth, proliferation, and angiogenesis of tumor cells. Therefore, inhibiting these metabolic pathways can be considered a potential therapeutic strategy in human malignancies. On the other hand, according to previous studies, pharmacological inhibiting of metabolic pathways using dual-pathway inhibitors can considerably inhibit tumor growth and progression, more than suppressing each pathway separately. This review aims to summarize the latest metabolic interventions by dual pathway inhibitors and discuss the achievements and limitations of this therapeutic tactic.

**Abstract:**

The metabolism of tumors and immune cells in the tumor microenvironment (TME) can affect the fate of cancer and immune responses. Metabolic reprogramming can occur following the activation of metabolic-related signaling pathways, such as phosphoinositide 3-kinases (PI3Ks) and the mammalian target of rapamycin (mTOR). Moreover, various tumor-derived immunosuppressive metabolites following metabolic reprogramming also affect antitumor immune responses. Evidence shows that intervention in the metabolic pathways of tumors or immune cells can be an attractive and novel treatment option for cancer. For instance, administrating inhibitors of various signaling pathways, such as phosphoinositide 3-kinases (PI3Ks), can improve T cell-mediated antitumor immune responses. However, dual pathway inhibitors can significantly suppress tumor growth more than they inhibit each pathway separately. This review discusses the latest metabolic interventions by dual pathway inhibitors as well as the advantages and disadvantages of this therapeutic approach.

## 1. Introduction

Metabolic processes turn nutrients into molecules termed metabolites through a complex network of biochemical reactions, generating energy, redox equivalents, and macromolecules, such as RNA, DNA, proteins, and lipids essential for cell functions and survival [1,2]. Cytosolic glycolysis under anaerobic conditions and mitochondrial oxidative phosphorylation under aerobic conditions are energy sources for normal cells, respectively [3]. In contrast, according to the “Warburg effect”, cancerous cells desire to obtain energy via cytosolic glycolysis than oxidative phosphorylation, even under aerobic conditions [4,5]. Following activation of glycolysis, glycolytic tumor cells produce lactate, which is considered an energetic fuel for oxidative tumor cells. Monocarboxylate transporters (MCTs) catalyze the proton-linked transport of lactate and other monocarboxylates across cell membranes [6] (Figure 1). The justification for this tendency of tumor cells is their uncontrollable proliferation and the need for a fast ATP supply that is only accessible via glycolysis [7,8]. On the other hand, various principal metabolism pathways can be dysregulated in tumor cells [1]. According to available knowledge, immune responses are associated with significant changes in tissue metabolism, such as nutrient depletion, consumption of oxygen, and the generation of reactive oxygen and nitrogen intermediates [9,10,11].

Moreover, in the TME, numerous metabolites can affect immune cells’ differentiation and effector function [12]. However, in the TME, there is always fierce competition between immune and tumor cells to consume nutrients, and tumor cells usually win this competition due to their proliferative power and aggressive characteristics [13]. Correspondingly, metabolic interventions may be a potential therapeutic approach for treating malignancies.

It has been revealed that various signaling pathways, such as mitogen-activated protein kinase (MAPK), AMP-activated protein kinase (AMPK), mammalian target of rapamycin (mTOR), hypoxia-inducible factor 1-alpha (HIF-1α), PI3K/AKT, Ras, and insulin receptor are involved in cell metabolism. Interestingly, these pathways and cross-regulation could affect tumor growth and T cell-mediated immunity [14,15]. In this regard, several studies showed that pharmacological intervention using various inhibitors of these pathways could determine T cells’ metabolic fitness and persistence of these immune cells [16]. For instance, sirolimus analogs such as mTOR inhibitors are now being studied in phase II and III clinical trials because mTOR signaling dysfunction induces cellular proliferation and has been associated with various human malignancies [17]. However, despite the benefits of this therapeutic method, using these inhibitors could have adverse reactions such as nephrotoxicity and increased risk of infections requiring conscientious monitoring of the treatment [18]. The PI3K is an essential mediator of tumor cell growth, proliferation, and survival because overactivated PI3K alpha (PI3KA) following tumor mutations is critical for downstream signals of receptor tyrosine. These data indicate that administering selective PI3KA inhibitors might be attractive therapeutic agents in cancer treatment. The mTOR is a PI3K downstream kinase pivotal in cell growth and metabolism. Therefore, inhibition of the mTOR is beneficial in clinical settings for several types of cancers [19].

Furthermore, dual pathway inhibitors could be more efficient than controlling the metabolic pathways separately. Simultaneous inhibition of glycolysis and oxidative phosphorylation, as well as PI3K/AKT/mTOR and other pathways and involved molecules with dual inhibitors, showed that this strategy is effective in most cases and helps to prevent the growth and development of the tumor [20,21,22,23]. However, this response to treatment can be different in different cancers.

This review summarized the metabolism of tumor and immune cells and their effect on each other. Furthermore, critical signaling pathways involved in tumor and immune cell metabolism, related therapeutic interventions with dual inhibitors but not dual inhibition of metabolic pathways with combination regimens, and the advantages and disadvantages of these dual inhibitors are discussed.

## 2. Metabolism of Tumor and Immune Cells

### 2.1. Tumor Cells

Due to the high proliferation rate of tumor cells, regardless of whether the condition is aerobic or anaerobic, cytosolic glycolysis is the preferred method of providing ATP to their growth [24]. Researchers have demonstrated that tumor cells generate pyruvate under hypoxic conditions via the glycolysis pathway, producing lactic acid by pyruvate kinase type M2 instead of entering the mitochondrial oxidative phosphorylation and acetyl-CoA formation [25]. The tumor cells also generate biological macromolecules to replicate themselves using serine metabolism and the pentose phosphate pathway (PPP) [26,27]. The environmental conditions and concentration of nutrients for tumor cells determine which path and which macromolecules they use to find the optimal conditions for their growth and development. Therefore, in addition to decomposing glucose, tumor cells can use other macromolecules, such as amino acids, lipids, and fatty acids, to produce energy and grow [28,29,30].

Interestingly, when the concentration of glucose or glutamine is low (nutrients deprivation), tumor cells induce c-Myc to promote their survival via regulation of metabolic enzyme expression in the serine synthesis pathway, including phosphoglycerate dehydrogenase (PHGDH), phosphoserine aminotransferase 1 (PSAT1), phosphoserine phosphatase (PSPH), activating the de novo serine synthesis and preserving redox homeostasis [31]. Moreover, under nutrient-deficient conditions, tumor cells are able to use acetoacetate to produce acetyl-CoA and fatty acids, which guarantee their survival [32,33,34]. Ketone body decomposition by tumor cells also generates metabolites that can enter the tricarboxylic acid cycle (TCA), providing ATP for their survival [30]. Cell cycle arrest, autophagy, anoikis, and entosis are four forms of anchorage-independent survival [35]. Recently an investigation reported that tumor cells prioritize glutamine-derived TCA energy metabolism over glycolysis to support ATP and suppress augmented oxidative stress by interacting with cysteine, preserving an anchorage-independent survival [36]. These findings indicate that depending on the different conditions governing the TME, tumor cells can intelligently provide their required energy through metabolic reprogramming and using different pathways to prolong their survival.

### 2.2. Immune Cells

In general, energy consumption in immune cells is different in active and inactive states. Moreover, like cancer cells, immune cells also use the metabolic pathways mentioned in the previous section [37]. Different metabolic patterns can affect immune cell differentiation. Previous studies demonstrated that M_1_ macrophages, activated neutrophils, and inducible nitric oxide synthase (iNOS)-expressed dendritic cells (DCs) mainly use glycolysis for their energy supply [38]. In the resting state, DCs prefer to use oxidative phosphorylation for energy supply, but activation of these cells is associated with increased glycolysis and lipid metabolism changes, affecting their function [39,40]. Furthermore, neutrophils use pentose phosphate and aerobic glycolysis pathways, and glycolysis is involved in regulating several neutrophil functions, such as chemotaxis and respiratory burst [41].

T cells play a unique role in anti-tumor defense among immune cells, and according to the various microenvironment signals, their phenotypes are metabolically different from other immune cells. Evidence demonstrated that the metabolic pattern of naïve and memory T cells is in a basic nutrient intake mode, glycolysis rate is decreased, proliferation is in a minimum state, and ATP supply mainly depends on oxidative phosphorylation [42]. While in pathologic conditions such as cancer, naïve T cells have to differentiate into effector T cells to defend against tumor cells, which require metabolic changes and increased proliferation. These metabolic alterations intensify the absorption of nutrients and glycolysis rate and increase the synthesis of essential macromolecules, such as nucleotides, proteins, and lipids. Simultaneous with these metabolic changes, mitochondrial oxygen consumption is condensed, inducing an effector T cell proliferation [2].

In contrast, regulatory T cells (Tregs) and M_2_ macrophages principally use oxidative phosphorylation from fatty acid oxidation (FAO) to provide the energy they need [43]. B cells are other immune cells that are involved in humeral immunity. It has been reported that activated B cells prefer to use glycolysis. However, following B cell activation by lipopolysaccharide (LPS) or other antigens, mitochondrial metabolism and glycolysis are boosted in these cells [44,45]. Recently, it has been revealed that the upregulation of oncogene c-Myc and increased glycolysis are critical for generating functional regulatory B cells (Bregs) [46].

### 2.3. Nutritional Competition between Tumor Cells and Immune System Cells

A significant challenge for antitumor immune responses is the competition between tumor cells and immune cells to take up glucose, amino acids, fatty acids, growth factors, and other metabolites in the TME. The expression of related transporters on the surface of these cells can also influence the fate of tumors and the immune system’s response [13]. The most critical nutrient consumed and absorbed by tumor cells is glucose, which also serves as an essential energy substance for the differentiation, activation, and function of infiltrated immune cells in the TME, such as tumor-infiltrating lymphocytes (TILs) [47,48,49]. Competitive uptake of glucose by tumor cells to suppress the function of TILs is one of the tumor escape and immunosuppressive mechanisms of cancer [50]. Moreover, increased glycolytic activities of tumor cells, and generated metabolites, such as lactate, can suppress glucose consumption by TILs, their exhaustion, and damage to their functions [51,52]. Additionally, tumor heterogenicity, high acidity, hypoxia, and high concentrations of lactate and ROS in the TME stimulate immune escape and cancer development [52]. Consequently, targeting various involved metabolic pathways affecting T cell-mediated antitumor responses could be a potential approach to overcome the destructive effects of metabolic competition between immune and tumor cells [53] (Figure 2).

## 3. The Most Important Metabolic Pathways in Cancer and Therapeutic Interventions

### 3.1. PI3K/AKT/mTOR Pathway

PI3K is known as a group of plasma membrane-related lipid kinases. These kinases comprise p55 (regulatory), p110 (catalytic), and p85 (regulatory) subunits [54]. PI3K is categorized into PI3KI, PI3KII, and PI3KIII classes based on various structures and substrates [55]. The p85 regulatory subunit can bind and integrate signals from protein kinase C (PKC), tyrosine kinase-linked receptors, hormonal receptors, Src homology 2 domain-containing protein tyrosine phosphatase 1 (SHP1), Src, mutated Ras, Rac, and Rho, activating the p110 catalytic subunit and other downstream molecules [56]. Stabilizing the p110 subunit depends on its dimerization with the p85 subunit. As extracellular stimuli, hormones, cytokines, and growth factors activate PI3K in normal and physiologic conditions [57]. Activated PI3K induces the phosphorylation of phosphatidylinositol 4,5-bisphosphate to produce phosphatidylinositol 3,4,5-trisphosphate (PIP3), stimulating downstream kinases, such as AKT and 3-phosphoinositide-dependent protein kinase-1 (PDK1), and inducing cell growth and cell survival pathways [58,59]. It has been revealed that phosphatase and tensin homolog (PTEN) regulates the PI3K pathway via dephosphorylation of PIP3 to PIP2, inhibiting downstream kinase activation [56].

One of the leading downstream PI3K signaling effectors is mTOR, a serine/threonine protein kinase that regulates cell growth, proliferation, and metabolism [60,61]. Based on available knowledge, mTOR complex 1 (mTORC1) and mTOR complex 2 (mTORC2) are two structures of mTOR. These complexes have different functions; for instance, mTORC1 induces cell anabolism by promoting the synthesis of nucleic acid and protein while preventing cell catabolism-mediated processes such as autophagy. On the other hand, mTORC2 induces glutamine uptake via activating AGC kinases, resulting in the regulation of glutamine cell surface transporters [60]. Furthermore, mTORC1 induces glutamine synthesis by positively regulating glutamate dehydrogenase (GDH) and suppressing sirtuin 4 (SIRT4), which is responsible for GDH inhibition [62,63]. Since aerobic glycolysis is a hallmark of tumor cells, nitrogen and carbon are supplied by glutamine to facilitate anabolic processes and cell growth [64]. In tumor cells, it has been demonstrated that the mTOR pathway is responsible for stimulating tumorigenesis, inducing the expression of inhibitory molecules, such as programmed cell death ligand-1 (PDL-1), and suppressing anticancer immune responses [65].

In some human malignancies, mTOR gene mutations are reported because these malignancies can activate mTOR constitutively. According to the tumor genome sequencing datasets, thirty-three mTOR mutations involved in cancer have been identified. The discovered mutations are categorized into six distinct regions in the C-terminal half of mTOR. They are responsible for hindering the interactions between mTOR and DEP domain-containing mTOR-interacting protein (DEPTOR) (endogenous mTOR inhibitor), hyperactivating the mTOR pathway [66]. Other mutations are also related to mTORC1 and mTORC2-specific components and upstream elements, including oncogenes and tumor suppressors [67,68]. Moreover, several cancer-mediated mutations are reported in the PI3K pathway, the upstream of mTORC1 and mTORC2 [69]. For instance, mutations in *PIK3CA*, which encodes the p110α PI3K catalytic subunit, have been reported in several human malignancies, such as prostate, breast, endometrium, colon, and upper aerodigestive tract cancers [70].

As discussed, cancer cells require metabolic reprogramming to facilitate their proliferation, growth, biological functions, and survival. In this context, mTOR plays a regulatory role in cellular metabolism via upregulating the expression of ribosomal protein S6 kinase beta-1 (S6K1) and eukaryotic translation initiation factor 4E (eIF4E)-binding protein 1 (4E-BP1) [71]. In addition, the proliferation and growth of tumor cells are supported by mTOR-enhancing glucose metabolism by upregulating the transporter 1 (GlUT1), HIF1-α, and c-MYC, resulting in the enhancement of glycolytic enzymes, such as enolase (ENO), phosphofructokinase (PFK), and phosphoglucoisomerase (PGI) [72,73,74]. The signaling of mTORC1 and mTORC2 induces fatty acid uptake and lipogenesis to support tumor cell proliferation [74]. These complexes induce sterol regulatory element-binding protein 1 (SREBP-1) and the peroxisome proliferator-activated receptor γ (PPARγ), which are involved in promoting the expression of lipid and cholesterol homeostasis-associated enzymes, such as fatty acid transporter CD36, acetyl-CoA carboxylase 1 (ACC1), ATP citrate lyase (ACLY), and fatty acid synthase (FASN) [75,76,77]. It has been revealed that inhibiting the rapamycin-insensitive companion of the mammalian target of rapamycin (RICTOR) as a mTORC2 component, as well as inhibition of mTORC1, mTORC2, and PI3K, could remarkably interrupt the progression of pancreatic cancer and prolong survival in end-stage of the tumor [78]. Furthermore, overexpression of RICTOR is associated with lymph node metastasis, tumor progression, and poor prognosis [79]. Employing kinase inhibitors or using RICTOR knockdown are other therapeutic approaches in mTORC2-targeted cancer therapy, leading to suppression of tumor cell growth, migration, and metastasis [80,81]. In colorectal cancer (CRC), RICTOR deficiency could significantly decrease the pAktSer^473^ level and reduce the proliferation and growth of CRC cells [82]. AKT hyperactivation is another consequence of RICTOR upregulation, progressing tumor cells and decreasing overall survival. In human epidermal growth factor receptor 2 (EGFR2) positive breast cancer, the effectiveness of HER2/EGFR tyrosine kinase inhibitors such as lapatinib is increased following the knockdown of RICTOR or using kinase inhibitors [68].

According to the available evidence, it regulates the immune system components, including immune cell metabolism, differentiation, activation, effector function, and homeostasis in innate and adaptive immunity [83]. Moreover, the activation of PI3K/AKT/mTORC1 is essential for developing metabolic reprogramming effector CD4^+^ and CD8^+^ T cells [84,85]. Following the interaction of the T cell receptor (TCR) and the presented antigens, downstream signals sent by TCR, co-stimulatory molecules in immunologic synapses, as well as cytokine-mediated signals received by mTORC1 and mTORC2 and their complexes regulate the immune receptor pathways, transcription factors, migration, and metabolic reprogramming. In addition, mTOR signals are involved in determining the fate of T cells and which phenotype will be formed in them and go towards memory, regulatory, or effector T cells [85]. In this regard, an investigation demonstrated that T cells with Rheb deficiency could not differentiate into T helper 1 (Th1) and Th17 and generate related immune responses. In contrast, these T cells tend to differentiate into Th2 [86].

Interestingly, targeting mTORC2 signals through the knockdown of RICTOR in T cells prevents their differentiation into Th2 and enhances differentiation into Th1 and Th17 cells. Furthermore, the generation of Tregs depends on the selective deletion of mTORC1 and mTORC2 signals regardless of the existence of the exogenous transforming growth factor-beta (TGF-β) [86]. Therefore, rapamycin, as an mTOR inhibitor, can repress the activation and proliferation of T cells [87]. An experimental study showed that metabolic manipulation of naïve T cells and TILs during their expansion in vitro using Akt inhibitor VIII could induce the differentiation of T cells into memory T cells with suitable antitumor activity following reinfusion of these T cells to immune-deficient mice with multiple myeloma [88].

The metabolic interventions using pharmaceutic agents can affect metabolic fitness and T cell persistence [16]. An investigation on CD33-specific chimeric antigen receptor (CAR)-T cells showed that treating these engineered cells with LY294002, a PI3K inhibitor, in vitro led to less differentiation of these cells into shorter-lived effector forms with enhanced antitumor activity and persistence in mice. Inhibition of PI3K/AKT/mTOR was also associated with rising glycolytic flux following the activation of CAR-T cells [89]. In these CAR-T cells, using various co-stimulatory domains such as CD28 or 4-1BB could affect T cell metabolism and persistence. For instance, 4-1BB could induce mitochondrial biogenesis, oxidative phosphorylation, and differentiation into memory T cells, along with more in vivo persistence of T cells, while employing CD28 was associated with increasing glycolysis and effector differentiation of T cells [90]. These findings demonstrate that metabolic interventions could be related to improving the effectiveness of cell therapy in cancer; however, due to the metabolic alteration of T cells, it is possible to change the function and phenotype, and this type of intervention needs more studies.

### 3.2. AMPK Pathway

AMPK is considered a crucial molecule in regulating cell energy homeostasis by monitoring AMP, ADP, and ATP levels. AMPK comprises three subunits: α subunit (catalytic) and β and γ (regulatory) subunits and several tissues/organisms-specific isoforms, including α1, α2, β1, β2, γ1, γ2, γ3 [91]. Intracellular calcium ions through calcium/calmodulin-dependent protein kinase kinase 2 (CAMKK2) and adenine nucleotides are able to activate the AMPK pathway [92]. In stress conditions, including hypoxia, low glucose concentrations, and ischemia associated with ATP depletion, the AMPK pathway is also activated. This activation is regulated by cellular AMP/ADP/ATP that competitively binds to the γ subunit. These occurrences can stimulate Thr172 phosphorylation on the α subunit via the tumor suppressor liver kinase B1 (LKB1) or suppress Thr172 phosphorylation through dephosphorylating α subunit by phosphatases [93,94]. AMPK can also be suppressed by fructose 1,6-bisphosphate (FBP), a glucose metabolite [91]. Activating the AMPK can induce autophagy and fatty acid oxidation to supply and reload the intracellular ATP [95]. Since gluconeogenesis, protein, and lipid synthesis are ATP-consuming, AMPK negatively regulates biosynthetic processes to preserve ATP and control energy metabolism, activating immune cells [96]. These findings indicate that the AMPK pathway controls the balance between immune responses and energy metabolism [2]. On the other hand, AMPK activation inhibits various immune signaling pathways involved in the proliferation and activation of immunosuppressive immune cells, such as myeloid-derived suppressor cells (MDSCs) [96]. Accordingly, the AMPK pathway, as a metabolic regulator, may play an antitumoral role in cancer. In contrast, other studies showed that AMPK activation could be associated with the suppression of pro-inflammatory pathways, such as NFκB, and the differentiation of macrophages from the M1 into the M2 phenotype, enhancing the expression of anti-inflammatory cytokines, such as IL-10 [97,98]. Activation of the AMPK pathway via controlling energy metabolism is involved in the differentiation of T cells, affecting the function of these immune cells [2].

### 3.3. Adenosine Pathway

Following tissue injury or hypoxic TME, nucleoside adenosine levels are significantly amplified and bind to adenosine 2A receptor (A2AR) on the cell surfaces, inhibiting cytotoxic T cell/natural killer cells (NK) cell-mediated antitumor immune responses. CD73 and CD39 regulate the production of adenosine via the catabolism of ATP. CD39 converts ATP to AMP, and CD73 converts AMP to adenosine [99]. Immunosuppressive cells such as Tregs can express CD39, and activation of the A2AR pathway in these immune cells leads to the downregulation of inflammatory mediators and upregulation of anti-inflammatory mediators, such as IL-10, resulting in dephosphorylation of signal transducer and activator of transcription 5 (STAT5), inhibiting the NFκB pathway, and reducing IL-2R-mediated signals in T cells. Tregs generate adenosine through the co-expression of CD39/CD73, activating the adenosine pathway and overexpressing prostaglandin E2 (PGE2) receptor, EP2 receptors (EP2R) on the surface of responder T cells. In addition, adenylate cyclase activity increased following the adenosine pathway’s activation, leading to increased cAMP and promoting immunosuppressive responses [100].

## 4. Dual Pathway Inhibitors

So far, numerous studies have been performed on metabolic pathway inhibitors in cancer therapy, and relatively satisfactory outcomes have been achieved. However, there is also a theory that utilizing dual pathway inhibitors increases the effectiveness of cancer therapy. This section discusses the properties of these dual inhibitors and the consequences of their use in cancer treatment (Table 1). The chemical structure and molecular formula of dual inhibitors are also shown in Table 2.

### 4.1. Dual PI3K/AKT/mTOR Inhibitors

PI3K and mTOR belong to the family of phosphatidylinositol 3-kinase-related kinases (PIKKs). According to the structural and functional similarities of PI3K and mTOR, as well as studies on mTOR inhibitors, researchers synthesized inhibitors with dual functions, suppressing both PI3K and mTOR [143].

#### 4.1.1. Dactolisib

Dactolisib (BEZ235) is an imidazoquinoline targeting the PI3K and the mTOR, with robust antitumor activity. Dactolisib suppresses PI3K kinase and mTOR kinase in the PI3K/AKT/mTOR kinase pathway, inducing tumor cell apoptosis and inhibiting growth in PI3K/mTOR highly expressing cancer cells. In addition to causing tumor cell growth, proliferation, and survival, the PI3K/mTOR pathway also plays a crucial role in making the tumor resistant to conventional therapies, such as radiotherapy and chemotherapy [101].

It was investigated in non-small-cell lung cancer (NSCLC) cells with various EGFR statuses whether co-inhibiting PI3K and mTOR would improve the therapeutic outcomes. This study reported that BEZ235 repressed tumor growth in vitro and in vivo via promoting cell-cycle arrest at the G1 phase and reducing the cyclin D1/D3 expression. Additionally, BEZ235 synergistically promoted cisplatin-mediated apoptosis in NSCLC cells by boosting or persisting DNA damage. These data indicate that dual PI3K/mTOR inhibition by BEZ235 can be a potential anticancer agent that induces the efficacy of targeted therapy or chemotherapy [102].

An investigation on mantle cell lymphoma (MCL) cells showed that compared with everolimus (an mTOR inhibitor) or NVP-BKM120 (a PI3K inhibitor), BEZ235 could be more potent in suppressing the PI3K/Akt/mTOR pathway. Furthermore, BEZ235 can inhibit angiogenesis, migration, and invasion of tumor cells. Besides, it has been revealed that interleukin-4 (IL-4) and IL-6/signal transducer and activator of transcription 3 (STAT3) pathway are involved in chemoresistance. Regarding the role of IL-6 in inducing chemoresistance, it has been revealed that IL-6-mediated stem cell expansion and epithelial-mesenchymal transition (EMT) could be involved in this obstacle. Mechanistically, IL-6 induces upregulation of multidrug-resistant associated mediators, such as MDR1 and glutathione S transferase pi (GSTpi). Moreover, IL-6 protects tumor cells from the paclitaxel- and cisplatin-associated cytotoxic effects by downregulating caspase3 (Cas3) and upregulating antiapoptotic proteins, such as X-linked inhibitor of apoptosis (XIAP), B-cell lymphoma 2 (Bcl-2), and B-cell lymphoma-extra large (Bcl-xL) in resistant cancer cells. Further, IL-6 can induce activation of the PI3K/AKT pathway in resistant tumor cells [144]. There is no clear indication of the exact mechanism by which IL-4 contributes to chemoresistance in tumors; however, the evidence demonstrates that, similar to IL-6, IL-4 can regulate key antiapoptotic factors that may have functional effects on the chemoresistance [145].

Unlike everolimus and NVP-BKM120, BEZ235 can inhibit signals of these cytokines, improving the effectiveness of chemotherapy [103]. These findings indicate that dual pathway inhibitors can be more effective than single pathway inhibition, inhibiting the PI3K/Akt/mTOR pathway at multiple levels. Combining BEZ235 with dexamethasone in acute lymphoblastic leukemia (ALL) showed that along with inhibiting the PI3K/AKT/mTOR pathway, antileukemic effects of dexamethasone were improved in vitro and in vivo. AKT1 is responsible for repressing dexamethasone-induced tumor cell apoptosis. Therefore, BEZ235, by inhibiting AKT and downregulating myeloid cell leukemia-1 (MCL-1), can induce dexamethasone-mediated apoptotic pathways in malignant cells [104]. A phase Ib dose-escalation clinical trial demonstrated that combining everolimus and BEZ235 (orally in escalating doses of 200, 400, and 800 mg/day plus everolimus at 2.5 mg/day in 28-day cycles) and this therapeutic regimen was associated with poor efficacy and tolerance. The remarkable feature of BEZ235 administration was that its oral administration could not be a suitable option for treatment due to low bioavailability and gastrointestinal toxicity.

In contrast, the systemic administration of this inhibitor can have better effectiveness in a dose-dependent manner [146]. Another phase I/Ib, multicenter, open-label by administrating different doses of BEZ235 to patients with HER2^+^ breast cancer showed that the effect of this drug was partially observed in only 13% of patients. Side effects, including nausea, diarrhea, and vomiting, were reported in patients. Moreover, BEZ235 showed more variability and effect in doses higher than 100 mg, although high doses were associated with gastrointestinal toxicity [105].

On the other hand, patients with advanced pancreatic neuroendocrine tumors (pNET) were treated with oral everolimus 10 mg once daily or oral BEZ235 400 mg twice daily on a continuous dosing schedule. Findings showed that the median progression-free survival (PFS) in BEZ235 treated group was 8.2 months versus 10.8 months in patients treated with everolimus. The most frequent adverse effects in patients with BEZ235 were diarrhea, stomatitis, and nausea. These results show that BEZ235 cannot be more effective than everolimus, at least in terms of PFS. On the other hand, this dual inhibitor’s side effects are more than everolimus. However, this response to treatment may change in cancers and patients with different conditions [147].

#### 4.1.2. Gedatolisib

Gedatolisib (PKI-587) is a dual inhibitor targeting the PI3K and mTOR kinases in the PI3K/mTOR signaling pathway, with potential antitumor activity. Evidence demonstrated that following intravenous administration of gedatolisib, it inhibits both mTOR and PI3K kinases, inducing apoptosis and suppressing the growth of tumor cells overexpressing PI3K/mTOR. Moreover, gedatolisib can enhance radio and chemosensitivity by inhibiting the PI3K/AKT/mTOR pathways to decrease DNA damage repair mechanisms [106]. Recently, an investigation reported that combining PKI-587 with Cofetuzumab Pelidotin, a protein tyrosine kinase 7 (PTK7)-targeted, auristatin-based antibody-drug conjugate in patients with metastatic triple-negative breast cancer (TNBC) was associated with promising clinical activity, two months median PFS, and moderate toxicity (anorexia nausea, mucositis, and fatigue) [107]. PKI-587 can increase radiosensitization. A study showed that DNA damage was increased in SK-Hep1 xenograft hepatocellular carcinoma (HCC) models, combining ionizing radiation with PKI-587, and G_0_/G_1_ cell-cycle arrest, as well as apoptosis, were induced in tumor cells. Accordingly, repressing the PI3K/AKT/mTOR and DNA damage repair pathways by PKI-587 can stimulate the radiosensitizing of HCC cells [108]. The prognosis in T-cell ALL patients (T-ALL) is poor. Changes in the PI3K/mTOR signaling pathway are responsible for relapse and treatment failure because the PI3K/mTOR pathway is overactivated in relapsed T-ALL patients. This study demonstrated that PKI-587 inhibited T-ALL cell line proliferation and colony formation via selective suppression of the PI3K/mTOR pathway without disturbing the mitogen-activated protein kinase (MAPK) pathway in vitro and in vivo. Furthermore, PKI-587 reduces tumor load and progression, lengthening survival rates in immune-deficient mice xenograft models without causing weight loss in the mice treated with the inhibitor [109]. It seems that PKI-587 can be a suitable option for treating human malignancies. However, combination therapy using PKI-587 can increase the effectiveness of the treatment by creating synergistic responses.

#### 4.1.3. Voxtalisib

Voxtalisib (SAR245409) is a powerful class-I PI3Ks, mTORC1, and mTORC2 inhibitor [148]. It has been reported that voxtalisib could suppress the phosphorylation of PI3K and control the incorporation of mTOR effector in cancer cells [149]. In a phase Ib clinical trial on patients with advanced malignant tumors, 90 mg pimasertib (a MEK1/2 inhibitor) and 70 mg voxtalisib was administrated, and findings showed that this combination regimen was not well tolerated and did not have a significant effect on the survival of patients with advanced solid tumor. The most commonly observed adverse events in this study were diarrhea, nausea, and fatigue [110]. It appears that patient drug tolerance depends on voxtalisib dose and schedule. A phase I clinical trial administrated a combination of voxtalisib with temozolomide, with or without radiation therapy, to patients with high-grade glioma. Outcomes showed that the maximum tolerated doses (MTDs) for voxtalisib in combination with temozolomide were 90 mg once a day and 40 mg twice daily. The most frequently experienced adverse events in this study were nausea, fatigue, thrombocytopenia, diarrhea, and lymphopenia. This study showed that voxtalisib, combined with temozolomide with or without radiation therapy, could effectively treat high-grade gliomas with acceptable safety [111].

#### 4.1.4. Bimiralisib

Bimiralisib (PQR309) is known as a pan-class I PI3K/mTOR antagonist that vigorously represses PI3Kα and mTOR. According to the biochemical experiments, bimiralisib has less influence on PI3Kβ and cannot remarkably inhibit other protein kinases [150]. It has been revealed that the PI3K/mTOR pathway is involved in several lymphoma types. Therefore, pharmacologic inhibition of this pathway may benefit patients with lymphoma. A preclinical lymphoma model demonstrated that bimiralisib showed anti-lymphoma activity in vitro alone or combined with other anticancer drugs, such as panobinostat, venetoclax, lenalidomide, ibrutinib, ARV-825, rituximab, and marizomib. This study demonstrated that bimiralisib could induce the expression of *HRK*, *YPEL3*, and *TP63*, while the gene expression of *HSPA8* and *HSPA1B*, *CCDC86*, *PAK1IP1*, and *MIR155HG* was downregulated following treatment [112]. A dose-escalation, open-label phase I trial evaluated the anticancer effects and safety of bimiralisib (dose 10 to 150 mg) in patients with advanced solid tumors. Results showed that partial response was detectable following bimiralisib therapy in a patient with metastatic thymus malignancy.

Moreover, disease volume was reduced to one-quarter in a patient with sinonasal cancer, and a patient with clear cell Bartholin’s gland cancer experienced stable disease for more than sixteen weeks. The MTD and recommended phase 2 dose of bimiralisib was considered to be 80 mg orally once per day. Analysis of tumor biopsies revealed that bimiralisib exerts its antitumor effects by downregulating the PI3K pathway phosphoprotein. Additionally, common adverse events, including hyperglycemia, fatigue, nausea, constipation, diarrhea, rash, vomiting, and anorexia, were detected in about 30% of patients [113]. Interestingly, bimiralisib can effectively cross the brain–blood barrier (BBB) compared with BEZ235 and voxtalisib [112,114]. This feature of bimiralisib can facilitate its delivery to the tumor tissue in brain tumors and improve the effectiveness of the treatment.

#### 4.1.5. Paxalisib

Paxalisib (GDC-0084) is known as a selective and potent oral brain-penetrant dual inhibitor of PI3K and mTOR kinase. Paxalisib was exclusively designed for treating brain tumors, such as progressive or recurrent glioma, because it can efficiently cross the BBB to improve drug delivery to the brain. Experimental studies have shown that paxalisib could inhibit the growth of tumor cells in a dose-dependent manner [115,116,117]. Based on available knowledge, the PI3K/Akt/mTOR pathway is overactivated due to the *PIK3CA* mutations in up to 70% of brain metastases in patients with breast cancer. A preclinical study showed that paxalisib significantly reduced cell viability and phosphorylation of AKT and p70 S6 kinase.

Moreover, apoptosis of *PIK3CA*-mutant breast cancer brain metastatic cells was increased following the treatment in lines in a dose-dependent fashion [118]. Therefore, using paxalisib may be effective in brain cancers and brain metastatic cancers. However, this dual inhibitor can be effective in other malignancies, such as cutaneous squamous cell carcinoma (cSCC). In this context, an investigation reported that paxalisib treatment at nanomole doses potently repressed the proliferation and survival of SCC-13, SCL-1, and A431 cell lines as well as primary human cSCC cells via induction of apoptosis and cell cycle arrest in the cSCC cells. Interestingly, in addition to its more lethal effect on tumor cells than other PI3K-Akt-mTOR pathway inhibitors, paxalisib was non-toxic to normal skin cells, including keratinocytes and fibroblasts [119]. The mechanism of action of paxalisib is inhibiting the phosphorylation of fundamental components of the PI3K-Akt-mTOR pathway, such as Akt, S6, p85, and S6K1. Furthermore, paxalisib hinders the activation of DNA-PKcs in cSCC cells [119].

#### 4.1.6. Omipalisib

Omipalisib (GSK2126458) is an oral dual PI3K/mTOR inhibitor that suppresses the growth and progression of cancer cells [151]. It has been revealed that omipalisib treatment could prevent the colony formation of cancer stem cells and induce autophagic cell death because clonogenicity depends on basic fibroblast growth factor (bFGF) and Insulin-like growth factor 1 (IGF-1) signaling via AKT and ERK pathways and omipalisib in combination with an ERK inhibitor, such as MEK162 can suppress colony formation [121]. Anti-proliferative effect of omipalisib on AML cell lines was explored and revealed that omipalisib could considerably induce G_0_/G_1_ cell cycle arrest in OCI-AML3 HL60 and THP1 cell lines. As discussed, omipalisib downregulates the phosphorylation of mTOR, AKT, 4E-BP1, and S6K. Moreover, metabolic pathway enrichment analysis showed that metabolites related to amino acid metabolisms were remarkably decreased upon treatment with omipalisib.

Additionally, following treatment of the OCI-AML3 cells with omipalisib, the expression of several essential genes, including *PHGDH*, *PSPH*, *PSAT1*, *MTHFD1/2*, and *SHMT1/2,* in the glycine and serine synthesis pathway, were significantly downregulated in these cells. Due to reducing the energy levels, the biosynthesis and functions of the mitochondria could probably be affected by omipalisib [122]. Additionally, studies on mice models showed that 0.2 or 1 mg/kg oral administration of omipalisib could notably reduce tumor growth without apparent alteration in the body weight of treated animals [123].

#### 4.1.7. SF1126

SF1126 is an RGD-conjugated LY294002 pro-drug with high solubility and antiangiogenic properties that can bind to specific integrins in the TME [152]. Therefore, the administration of SF1126 enhances delivery to the TME and tumor vasculature. Recent studies have shown that this compound can inhibit PI3K/AKT/mTOR and bromodomain-containing protein 4 (BRD4) pathways in cancer cells [124,125]. A study treated CRC cell lines as well as primary human colon cancer cells isolated from human tumors with SF1126, and findings showed that this drug could inhibit tumor cell growth and induce apoptosis. SF1126 also could lead to cell cycle arrest in cancer cells [124]. Another study reported that SF1126 treatment revokes the HIF-2α stabilization in VHL-mutated RCC cell lines under normoxic and hypoxic conditions. In addition, SF1126 subcutaneously administration to RCC-xenografted mice remarkably inhibited angiogenesis, tumor growth, and progression. SF1126 also could suppress integrin-mediated tumor cell migration and block the integrin-induced guanosine diphosphate (GDP)-Rac family small GTPase 1 (Rac1) conversion to its active state [126].

#### 4.1.8. PF-04691502

PF-04691502 is another dual PI3K/mTOR inhibitor that can repress tumor growth and progression through the induction of apoptosis. PF-04691502 also improves the radiosensitivity of several human malignancies [127]. It has been reported that PF-04691502 could inhibit the growth, proliferation, migration, and invasion of bladder cancer cells. Additionally, it can enhance the apoptosis of these tumor cells through the intrinsic pathway. PF-04691502 reduces the expression of the PI3K/Akt/mTOR pathway and myeloid leukemia 1 (*MCL-1*) in bladder cancer cells. As with several of the dual inhibitors discussed, PF-04691502 can also increase the effectiveness of chemotherapy and increase tumor cells’ sensitivity to radiotherapy [128].

Advanced-stage gastroenteropancreatic neuroendocrine tumors (GEP-NETs) are associated with poor prognosis despite radiotherapy and chemotherapy. Treatment of NET cell lines (QGP-1 and BON) with PF-04691502 downregulated the expression of pAKT for up to 72 h than in the control group. Surprisingly, concurrent treatment with PF-04691502 and radiotherapy did not enhance apoptosis in NET cells, while adding PF-04691502 48 h upon radiotherapy considerably induced apoptosis compared to radiotherapy or PF-04691502 therapy alone [129]. These outcomes indicate that combining radiation and PF-04691502 could be a novel and potential therapeutic approach for treating NETs [153].

In patients with T-cell lymphomas (CTCLs) and Sézary syndrome (SS), overactivation of the PI3K/AKT/mTOR pathway is demonstrable. Therefore, blocking this pathway signifies a potential therapeutic option against the cutaneous CTCLs [130]. Treatment with PF-04691502 suppressed the growth of the CTCL cell lines and derived tumor cells from SS patients. The PF-04691502 induced the apoptotic cascades and G1 cell arrest in the cell cycle of CTCL cell lines, whereas, in SS patients, its action was primarily due to the induction of strong apoptosis. Notably, PF-04691502 only mildly affected healthy donors obtained T cells.

Moreover, PF-04691502 suppressed CXCL12-related cell recruitment and migration in all studied groups. Following the treatment, along with increased survival, it was revealed that tumor volume reduced from 936 mm^3^ in the control group to 400 mm^3^ in treated mice. Additionally, tumor weight was decreased from 0.56 g in controls to 0.2 g in treated mice [153].

#### 4.1.9. Samotolisib

Samotolisib (LY3023414) is an orally available dual kinase inhibitor of class I PI3K and mTOR [131]. Preclinical studies showed that combining samotolisib with prexasertib, a checkpoint kinase 1 inhibitor (samotolisib 200 mg orally twice daily plus prexasertib 105 mg/m^2^ intravenously every 14 days), could have anticancer activity in preclinical models and preliminary value in seriously pretreated patients; however, the clinical combination was accompanied by toxicity, which should be considered in future trials [131]. A double-blind, placebo-controlled phase Ib/II trial combined samotolisib with enzalutamide (a nonsteroidal antiandrogen medication used to treat prostate cancer) in patients with metastatic castration-resistant prostate cancer. This study showed that the combination of samotolisib with enzalutamide was well tolerated and pointedly improved PFS in studied patients [132]. Evidence demonstrated that fatigue, nausea, vomiting, and diarrhea were the most frequent adverse events following treatment with samotolisib [133]. In anal dysplasia and anal cancer, inhibition of the PI3K/AKT/mTOR pathway is a practical approach. In K14E6/E7 mice treated with topical samotolisib, squamous cell carcinoma was inhibited after 15 weeks of treatment initiation in a sex-dependent manner (only male mice) [134].

#### 4.1.10. PWT33597

PWT33597 is another dual kinase inhibitor that, based on biochemical assays, represses PI3K alpha and mTOR. PWT33597 profiling showed little or no cross-reactivity with protein kinases, including tyrosine kinases or serine/threonine [19]. Treatment of mutationally activated PI3K alpha in HCT116 and NCI-H460 tumor cells with PWT33597 showed that this drug could inhibit mTOR pathway proteins and PI3K. Moreover, PWT33597 exhibited promising pharmacokinetic properties in multiple tumor xenograft models via an enduring reserve of PI3K and mTOR pathway signaling [19]. Several drugs that inhibit mTORC1 (rapalogs) are approved for the treatment of advanced renal cell carcinoma (RCC) [154]. However, the effectiveness of these drugs is limited to a specific subset of patients and is not enduring. It is proposed to administer PWT33597 to renal xenograft models in which both mTORC1 and mTORC2 inhibitions and PI3K inhibition may increase the effectiveness of the treatment by directly targeting multiple signaling nodes, including the vascular endothelial growth factor receptors (VEGFRs). PWT33597 was tested in VHL^−/−^, PTEN^−/−^ xenografts compared to rapamycin as a mTORC1 inhibitor and sorafenib, a VEGFR/RAF inhibitor. The results showed that despite the tumor growth-inhibitory properties of sorafenib and rapamycin (64%), PWT33597 had a much higher growth-inhibitory effect (93%). PWT33597 was more efficient than paxalisib (a pan-PI3K inhibitor) in inhibiting tumor growth, significantly reducing tumor weight and size. Furthermore, PWT33597 increases cleaved caspase 3 (an apoptotic indicator) [135].

#### 4.1.11. Apitolisib

Apitolisib (GDC-0980) is a novel dual PI3K/mTOR inhibitor. Apitolisib treatment strongly reduced the phosphorylation of AKT and mTOR and decreased growth in two cholangiocarcinoma (*CCA*) cell lines, SNU1196 and SNU478. Apitolisib also improved the effects of chemotherapeutic agents, such as cisplatin or gemcitabine, in vitro and boosted the cleavage of PARP. Additionally, combining apitolisib with chemotherapy in a mouse xenograft model of CCA decreased colony formation by SNU1196 and SNU478 cells and inhibited tumor cell growth [136]. Dysregulated PI3K/AKT/mTOR signals are responsible for tumorigenesis via inducing tumor growth, metastasis, and resistance to antitumor therapies in glioblastoma. Therefore, this axis could be an attractive therapeutic target for pharmacological manipulation. Glioblastoma multiforme (GBM) cell lines (A-172 and U-118-MG) were treated with apitolisib, and the treatment was associated with time- and dose-dependent cytotoxicity and apoptosis. The action mechanism of apitolisib is probably the downregulation of protein kinase RNA-like endoplasmic reticulum kinase (PERK) expression, blocking its inhibitory effect on protein synthesis, intensifying translation, and inducing apoptosis [137]. In contrast, a randomized open-label phase II trial reported that due to adverse events, such as hyperglycemia and rash, apitolisib could not effectively treat metastatic RCC, compared with everolimus [155]. Probably, the effect of this inhibitor can be different in different cancers.

### 4.2. Other Potential Dual Inhibitors

A cancer therapeutic approach is the dual inhibition of critical metabolic pathways, such as glycolysis and oxidative phosphorylation, that breaks cancer cells’ metabolic plasticity and limits the provided energy supply [156,157]. In this regard, an aptamer-based artificial enzyme was designed and constructed by arginine aptamer-modified carbon-dots-doped graphitic carbon nitride (AptCCN) to inhibit glycolysis and oxidative phosphorylation concurrently. The dAptCCN is able to capture intracellular arginine and convert arginine to nitric oxide (NO) via oxidation under red light irradiation. Evidence showed that the depletion of arginine and NO stress suppress glycolysis and oxidative phosphorylation, blocking energy supply and inducing tumor cell apoptosis [138]. Numerous tumor cells have been shown to increase the expression of nicotinamide phosphoribosyltransferase (NAMPT), which is essential for NAD^+^ salvage.

Consequently, employing NAMPT inhibitors could be an attractive option for cancer therapy [158]. KPT-9274 is a dual NAMPT/p21-activated kinase 4 (PAK4)/inhibitor that decreases the NAD^+^/NADH ratio in cancer cells, inhibiting tumor growth in sarcoma mice models and RCC [139,159]. KPT-9274 also induces antitumor immune responses via improving tumor antigen presentation and rising interferons (IFN)-α and IFN-γ responses [139]. GMX1778 is another NAMPT inhibitor that was used in murine GMB by microparticles. A study on GBM models reported that combining immune checkpoint inhibitors with GMX1778 increased the survival of treated animals [160]. GMX1778 increases the expression of programmed cell death ligan-1 (PD-L1) via NAD^+^ depletion and induces recruiting effector immune cells, such as CD4^+^ and CD8^+^ T cells. The frequency of M2-macrophages as immunosuppressive cells also decreased following treatment with GMX1778.

As discussed, tumor cells are capable of glucose metabolic alteration from oxidative phosphorylation to cytoplasmic glycolysis; pyruvate dehydrogenase kinases (PDKs) and lactate dehydrogenase A (LDHA) are crucial enzymes in this occurrence. Therefore, inhibiting these enzymes might be a promising approach in cancer therapy. An investigation designed two PDK/LDHA inhibitors (20e and 20k) that could decrease lactate formation and enhance oxygen consumption in A549 cells. These data indicate that these inhibitors can regulate the glucose metabolic pathways in cancer cells [140].

Type II topoisomerases are responsible for changing DNA topology via generating transient DNA double-strand breaks and are crucial for eukaryotic cells [161]. It has been revealed that dual inhibitors of kinases and topoisomerases II could be a potential therapeutic approach in cancer therapy. Designing dual inhibitors may also be a valuable and exciting strategy to overcome resistance to topoisomerase-targeted drugs due to the structural similarities between the topoisomerase II and other proteins, such as heat shock protein 90 (Hsp90), which is involved in DNA repair mechanisms [162].

Lysine (K)-specific demethylase 1A (*KDM1A*) is a flavin-dependent amine oxidase that is involved in the demethylation of lysine 3 and 4 in histone 3 tails (H3K4 and H3K9) [163]. Evidence showed that the upregulation of KDM1A is associated with multiple human disorders, such as cancer, through reduced methylation at the H3K4 and H3K9. Moreover, the demethylation of H3K4 and H3K9 leads to the condensation of chromatin, suppressing the transcription of several anticancer gene regions, such as DNA methyltransferase-1 (DNMT-1), p53, p21, GATA-binding factor (GATA)-1 and GATA-2. Accordingly, KDM1A inhibition can be beneficial in suppressing tumors [141]. On the other hand, spermine oxidase (*SMOX*) is an amine oxidase that can convert spermine and spermidine to spermidine and putrescine via deaminating aminopropyl [164]. Spermine and spermidine are involved in cellular functions, such as gene expression control, scavenging reactive oxygen species (ROS), cell cycle regulation, DNA structure maintenance, and protein synthesis [165]. Interestingly, *SMOX* has considerable sequence homology to KDM1A, which facilitates the design of dual inhibitors for cancer therapy [142]. In this context, an investigation reported that 3,5-diamino-1,2,4-triazole analogs could be used for dual inhibition of KDM1A and SMOX to treat pancreatic cancer [141].

## 5. Advantages and Disadvantages of Dual Pathway Inhibitors in Cancer Therapy

Evidence demonstrated that multitarget inhibitors are a promising tool for treating complicated disorders due to the inherent redundancy and robustness of numerous biological networks and pathways. In parallel, designing multitarget inhibitors is challenging for medicinal chemists [166] (Figure 3). One of the critical metabolic pathways that have been studied more is the PI3K/AKT/mTOR pathway, and significant dual inhibitors have been designed to inhibit the kinases of this pathway. There is a high prevalence of dysregulation of the PI3K/AKT/mTOR signaling pathway among cancerous cells [167,168,169]. There are different classes of PI3K/AKT/mTOR inhibitors, including mTOR inhibitors, PI3K/AKT inhibitors, and dual PI3K/AKT/mTOR inhibitors. The rationale of PI3K/AKT/mTOR inhibitor development is the existence of a negative feedback loop of S6K1 because durable inhibition of mTOR promotes the activation of PI3K/AKT [170].

Consequently, to overcome this challenge, employing dual PI3K/AKT/mTOR inhibitors could be a potential therapeutic approach. Lack of appropriate biomarkers that predict the activity of PI3K/AKT/mTOR inhibitors is another limitation in the clinical development of these dual inhibitors. However, in some human cancers, such as breast cancer, *PIK3CA* mutation is considered a biomarker for predicting PI3K/AKT/mTOR pathway activity [171]. Furthermore, WNT/β-catenin pathway-mediated *PIK3CA* mutations may reduce the sensitivity of tumor cells to the dual PI3K/mTOR inhibitor [172].

Clinical trials reported that common toxicities of administered PI3K/AKT/mTOR inhibitors were rash, gastrointestinal adverse events, fatigue, and asthenia. Furthermore, due to the impact of PI3K signaling on glucose metabolism, hyperglycemia has also been variable [173]. However, other adverse events may also be reported following the administration of dual pathway inhibitors. The induction of RICTOR acetylation by glucose is another challenge in targeting the PI3K/AKT/mTOR pathway because it leads to the activation of mTORC2 and therapeutic resistance to PI3K/AKT inhibitors. In glioblastoma cells, overactivation of mTORC2 following glucose-mediated RICTOR acetylation promotes the epidermal growth factor receptor vIII (EGFRvIII) signaling [174]. Besides, it has been demonstrated that monotherapy with mTOR inhibitors, such as rapamycin, suppresses antitumor immune responses via inhibiting effector CD8^+^ T cells, increasing Tregs frequency, and modulating dendritic cells and antigen presentation [175]. Therefore, determining the exact role of the mTOR pathway in the microenvironment of different tumors plays an essential role in the success of treatment using PI3K/AKT/mTOR inhibitors. For instance, it has been recently stated that inhibiting the mTOR pathway significantly stimulates antitumor immune response via increasing the frequency of long-lived CD8^+^ memory T cells and improving tumor cell eradication [16]. Moreover, inhibition of the PI3K/AKT/mTOR pathway could be associated with reducing tumor cell growth, proliferation, migration, invasion, and survival. On the other hand, PI3K/AKT/mTOR inhibitors can improve tumor immunosurveillance efficacy by downregulating the immunosuppressive pathways and activating antitumor immune responses in the TME.

ATP-binding cassette (ABC) drug transporters, including ABCB1 and ABCG2, are involved in multidrug resistance [176]. It has been revealed that overexpression of these transporters reduced the efficacy of dual PI3K/AKT/mTOR inhibitors, such as LY3023414, in tumor cells. Since LY3023414 is a substrate for ABCB1 and ABCG2, these transporters, by their drug efflux function, significantly reduce intracellular levels of LY3023414 in tumor cells [177]. Moreover, pharmacokinetic alterations in PI3K/AKT/mTOR inhibitors should be noted in pharmacological interventions when the drugs are prescribed together. For instance, drug–drug interactions between these inhibitors, such as everolimus and BEZ235, can affect their steady-state pharmacokinetic parameters [146]. It is realized that everolimus is a substrate of the CYP3A4 enzyme as well as the P-glycoprotein (a drug transporter) enzymes. This drug is highly susceptible to any alterations in the level of the enzyme CYP3A [178]. Available metabolic-related findings demonstrate that BEZ235 may modulate the expression and activation of CYP3A4. It was hypothesized that everolimus and BEZ235 could interact due to their absorption, metabolism (pharmacokinetic properties), and pharmacodynamic pathways [179].

How the inhibitors are metabolized is also a critical issue in the effectiveness of the treatment. Some PI3K/AKT/mTOR dual inhibitors, such as PWT33597, are metabolized more slowly in vivo and interact less with cytochrome P450 enzyme, resulting in an enduring inhibition of the PI3K/AKT/mTOR pathway in xenograft tumors. However, PWT33597 administration in mice could be accompanied by transient increases in insulin plasma concentrations [19]. Therefore, considering a drug’s positive and negative aspects is critical in managing and increasing the success of cancer treatment with metabolic intervention.

## 6. Concluding Remarks

Pharmacological intervention in different metabolic pathways can lead to fundamental alterations in tumor cell metabolism and pathological function, affecting immune responses in the TME. The dual inhibitors of metabolic pathways can have a better effect in preventing the growth and progression of tumor cells due to the simultaneous inhibition of pathways such as the PI3K/AKT/mTOR pathway. However, in some cancers, such as advanced pancreatic neuroendocrine tumors (pNET), using inhibitors of each pathway separately has had a better effect than dual inhibitors. Despite the various advantages, administrating dual inhibitors has multiple challenges and limitations. For example, the mTOR pathway can sometimes trigger anti-tumor immune responses. In these cases, its inhibition may be associated with the suppression of the immune system, and this issue can entirely depend on the tumor type, signal, and stage. For instance, in melanoma, the PI3K/Akt, MyD88, and IKK pathways could be involved in IL-36β-mediated mTORC1 activation, promoting CD8^+^ T cell activation and inducing antitumor immune responses in vitro and in vivo [180]. Based on the available studies, it appears that combining dual inhibitors with other chemotherapeutic agents (paclitaxel and cisplatin) or other targeted therapies, such as trastuzumab or anti-immune checkpoint blockers (anti-PD-1 and anti-CTLA-4), can increase the effectiveness of the treatment [105,181,182]. However, common toxicities, especially gastrointestinal toxicities and drug dose adjustments, are also essential factors that should be considered in designing a pharmacologic protocol using monotherapy with dual inhibitors of metabolic pathways or combination therapies.

## Figures and Tables

**Figure 1 cancers-15-02043-f001:**
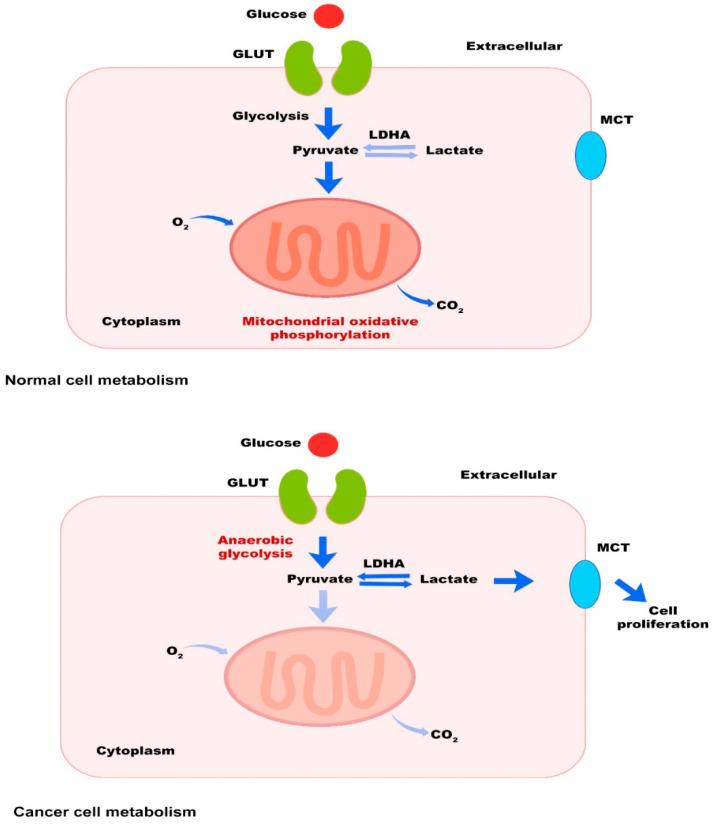
The Warburg effect. Most tumor cells produce energy, principally through glycolysis in the cytosol, producing lactic acid even in the presence of oxygen. MCTs catalyze the proton-linked transport of produced lactate across cell membranes. On the other hand, normal cells use oxidative phosphorylation in the mitochondria to produce energy under aerobic conditions.

**Figure 2 cancers-15-02043-f002:**
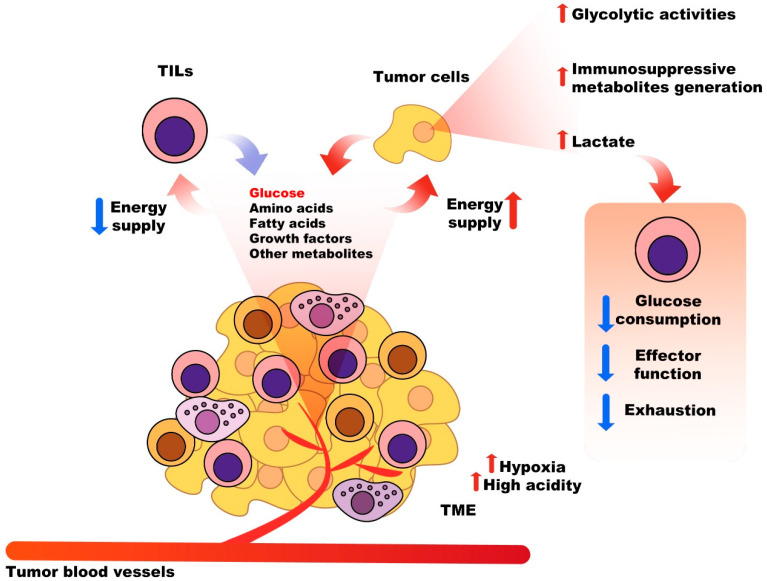
Metabolic competition between cancer cells and immune cells in the TME. There is a competition between tumor cells and immune cells to take up glucose, amino acids, fatty acids, growth factors, and other metabolites in the TME. The most critical nutrient consumed and absorbed by tumor cells is glucose, which also serves as an essential energy substance for the differentiation, activation, and function of infiltrated immune cells in the TME, such as TILs. Competitive uptake of glucose by tumor cells to suppress the function of TILs. Increased glycolytic activities of tumor cells, and generated metabolites, such as lactate, can suppress glucose consumption by TILs, and their exhaustion.

**Figure 3 cancers-15-02043-f003:**
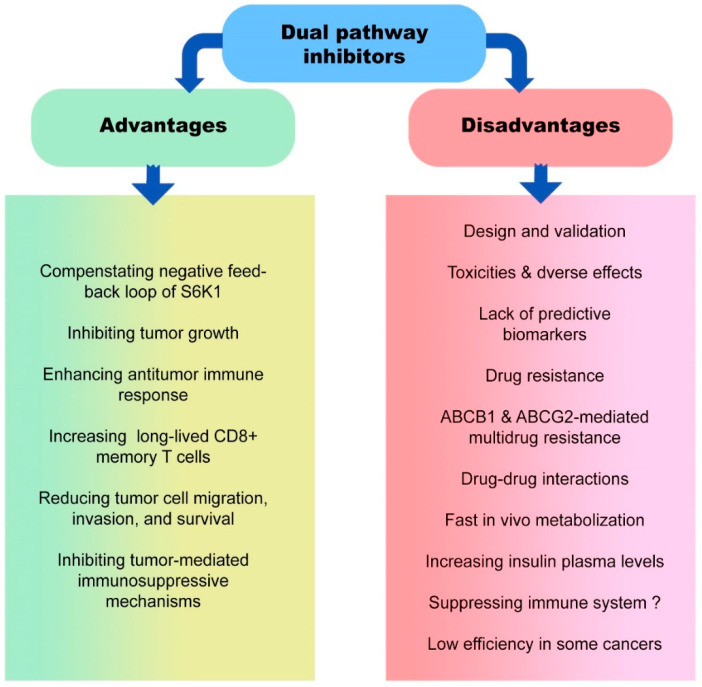
Advantages and disadvantages of using dual pathway inhibitors in cancer therapy.

**Table 1 cancers-15-02043-t001:** List of the most important dual pathway inhibitors.

Inhibitor	Target	Mechanism and Outcomes	Adverse Effects	Ref
Dactolisib (BEZ235)	PI3K/mTOR	⇑ G1 phase cell-cycle arrest ⇓ Cyclin D1/D3 expression ⇑ Tumor cell apoptosis ⇓ Tumor cell growth ⇓ Tumor cell migration ⇓ Chemoresistance ⇓ Angiogenesis ⇓ IL-4, IL-6, STAT3 ⇑ PFS	Nausea Diarrhea Vomiting Gastrointestinal toxicity Stomatitis	[101,102,103,104,105]
Gedatolisib (PKI-587)	PI3K/mTOR	⇑ G_0_/G_1_ cell-cycle arrest ⇑ Tumor cell apoptosis ⇓ Tumor cell growth ⇓ Chemoresistance ⇓ Colony formation ⇑ PFS	Anorexia Nausea Mucositis Fatigue	[106,107,108,109]
Voxtalisib (SAR245409)	Class-I PI3Ks, mTORC1/ mTORC2	⇑ Tumor cell apoptosis ⇓ Tumor cell growth ⇓ Chemoresistance	Diarrhea Nausea Fatigue	[110,111]
Bimiralisib (PQR309)	Pan-class I PI3K/mTOR	⇑ *HRK*, *YPEL3*, and *TP63* ⇓ *HSPA8*, *HSPA1B*, *CCDC86*, *PAK1IP1*, and *MIR155HG* ⇓ Tumor cell growth ⇑ Tumor cell apoptosis ⇓ Disease volume Cross the BBB ⇑ Drug delivery to the brain	Hyperglycemia Fatigue Nausea Constipation Diarrhea Rash Vomiting Anorexia	[112,113,114]
Paxalisib (GDC-0084)	PI3K/mTOR	⇓ Cell viability ⇓ The phosphorylation of AKT and p70 S6 kinase AKT, p85, and S6K1 ⇓ Tumor cell growth ⇑ Tumor cell apoptosis ⇑ Cell cycle arrest Cross the BBB ⇑ Drug delivery to the brain ⇓ DNA-PKcs	Rash Neutropenia Hyperglycemia	[115,116,117,118,119,120]
Omipalisib (GSK2126458)	PI3K/mTOR	⇓ Colony formation of cancer stem cells ⇑ Autophagic cell death ⇓ The phosphorylation of mTOR, AKT, 4E-BP1, and S6K ⇓ Tumor cell growth ⇑ Tumor cell apoptosis ⇑ G_0_/G_1_ cell cycle arrest ⇓ Amino acid metabolism ⇑ *PHGDH*, *PSPH*, *PSAT1*, *MTHFD1/2*, and *SHMT1/2*	-	[121,122,123]
SF1126	PI3K/mTOR BRD4	Binding to specific integrins in the TME ⇓ HIF-2α stabilization ⇓ Tumor cell growth ⇑ Tumor cell apoptosis ⇑ G_0_/G_1_ cell cycle arrest ⇓ Tumor cell migration ⇓ Angiogenesis	-	[124,125,126]
PF-04691502	PI3K/mTOR	⇑ Radiosensitivity of tumor cells ⇑ Effectiveness of chemotherapy ⇑ Tumor cell apoptosis ⇓ Tumor cell growth ⇓ Tumor cell migration ⇓ Expression of pAKT ⇑ G1 phase cell-cycle arrest ⇓ CXCL12-related cell recruitment and migration	-	[127,128,129,130]
Samotolisib (LY3023414)	PI3K/mTOR	⇑ Tumor cell apoptosis ⇓ Tumor cell growth ⇑ PFS	Fatigue Nausea Vomiting Diarrhea	[131,132,133,134]
PWT33597	PI3K/mTORC1 and mTORC2	⇓ Tumor growth ⇓ Tumor weight and size ⇑ Cleaved caspase 3	-	[135]
Apitolisib (GDC-0980)	PI3K/mTOR	⇑ Effects of cisplatin or gemcitabine ⇓ Tumor cell colony formation ⇑ Tumor cell apoptosis ⇓ Tumor cell growth and metastasis ⇓ PERK	Hyperglycemia Rash	[136,137]
AptCCN	Glycolysis & oxidative phosphorylation	⇑ Depletion of arginine ⇑ NO stress ⇓ Tumor growth	-	[138]
KPT-9274	NAMPT/p21 PAK4	⇓ NAD^+^/NADH ratio in cancer cells ⇓ Tumor growth ⇑ Antitumor immune responses ⇑ Tumor antigen presentation ⇑ IFN-α & IFN-γ	-	[139]
20e and 20k	PDK/LDHA	⇓ Lactate formation ⇑ Enhance oxygen consumption	-	[140]
3,5-diamino-1,2,4-triazole analogs	KDM1A/SMOX	⇓ demethylation of H3K4 and H3K9 ⇓ The condensation of chromatin ⇑ The transcription DNMT-1, p53, p21, GATA-1 & GATA-2 ⇑ Gene expression control ⇑ ROS ⇑ Cell cycle regulation ⇑ DNA structure maintenance	-	[141,142]

PI3K, phosphatidylinositol 3-kinase; mTOR, mammalian target of rapamycin; STAT3, signal transducer and activator of transcription 3; PFS, progression-free survival; BBB, blood–brain barrier; DNA-PKcs, DNA-dependent protein kinase catalytic subunit; HIF, hypoxia-inducible factor; 4E-BP1, eukaryotic initiation factor 4E-binding protein 1; PERK, protein kinase R (PKR)-like endoplasmic reticulum kinase; NO, nitric oxide; IFN, interferon; DNMT1, DNA-methyltransferase 1; ROS, reactive oxygen species.

**Table 2 cancers-15-02043-t002:** Chemical structure of dual pathway inhibitors.

Name	Molecular Formula	2D Structure
Dactolisib	C_30_H_23_N_5_O	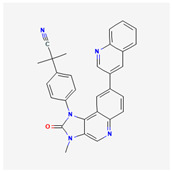
Gedatolisib	C_32_H_41_N_9_O_4_	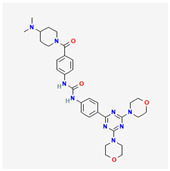
Voxtalisib	C_13_H_14_N_6_O	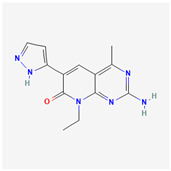
Bimiralisib	C_17_H_20_F_3_N_7_O_2_	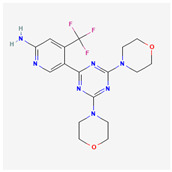
Paxalisib	C_18_H_22_N_8_O_2_	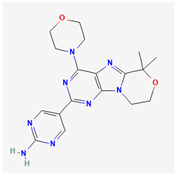
Omipalisib	C_25_H_17_F_2_N_5_O_3_S	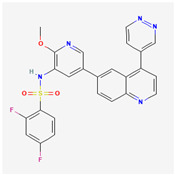
SF1126	C_39_H_48_N_8_O_14_	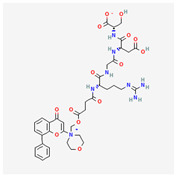
PF-04691502	C_22_H_27_N_5_O_4_	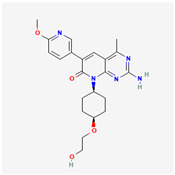
Samotolisib	C_22_H_27_N_5_O_4_	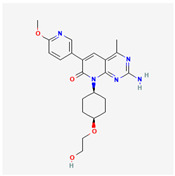
Apitolisib	C_23_H_30_N_8_O_3_S	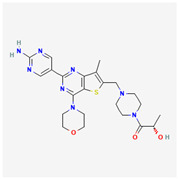
KPT-9274	C_35_H_29_F_3_N_43_	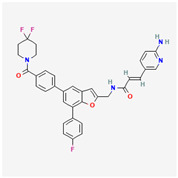

Figure source: PubChem^®^.

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
