# Peer review of "Metabolic Interventions in Tumor Immunity: Focus on Dual Pathway Inhibitors"

_cancers, 2023, doi:10.3390/cancers15072043_

Round 1
Reviewer 1 Report
The authors tried to discuss metabolic interventions using dual pathway inhibitors in cancer therapy. The authors have taken a new look at cancer metabolism and treatment using dual inhibitors of metabolic pathways. Despite the attractiveness of the topic, some concerns need to be addressed.
1. The Abstract should be revised according to the title.
ï¼’. In Figure 1, what is the role of MCT? Please explain in the text and Figure legend.
3. In section 3, It seems that more explanation should be given about the PI3K pathway.
ï¼”. Please add more data about drug-drug interactions in section 5
5. How do cytokines like IL-4 and IL-6 participate in chemoresistance? Please add more data to the text.
ï¼–. Please polish the text in terms of grammar and English writing.
Author Response
#Reveiwer1
The authors tried to discuss metabolic interventions using dual pathway inhibitors in cancer therapy. The authors have taken a new look at cancer metabolism and treatment using dual inhibitors of metabolic pathways. Despite the attractiveness of the topic, some concerns need to be addressed.
- The Abstract should be revised according to the title.
#Author’s response: Thank you so much for your time and proposal. The abstract has been revised.
- In Figure 1, what is the role of MCT? Please explain in the text and Figure legend.
#Author’s response: Thank you for your valuable comment. The monocarboxylate transporter 1 (MCT1) promotes lactate extrusion from the cytosol. The caption of the Figure has been revised.
- In section 3, It seems that more explanation should be given about the PI3K pathway.
#Author’s response: Related data about the PI3K pathway have been added to 3.1 subsection.
- Please add more data about drug-drug interactions in section 5
#Author’s response: Related data about drug-drug interactions have been added to section 5.
- How do cytokines like IL-4 and IL-6 participate in chemoresistance? Please add more data to the text.
#Author’s response: Thank you for your careful work. Regarding the role of IL-6 in inducing chemoresistance, it has been revealed that IL-6-mediated stem cell expansion and epithelial-mesenchymal transition (EMT) could be involved in this obstacle. Mechanistically, IL-6 induces upregulation of multidrug-resistant associated mediators, such as MDR1 and glutathione S transferase pi (GSTpi). Moreover, IL-6 protects tumor cells from the paclitaxel- and cisplatin-associated cytotoxic effects by downregulating caspase3 (Cas3) and upregulating antiapoptotic proteins, such as X-linked inhibitor of apoptosis (XIAP), B-cell lymphoma 2 (Bcl-2), and B-cell lymphoma-extra large (Bcl-xL) in resistant cancer cells. Further, IL-6 can induce activating the PI3K/AKT pathway in resistant tumor cells [1]. There is no clear indication of the exact mechanism by which IL-4 contributes to chemoresistance in tumors; however, the evidence demonstrates that, similar to IL-6, IL-4 can regulate key antiapoptotic factors that may have functional effects on the chemoresistance [2]. These findings have been added to the text.
- Please polish the text in terms of grammar and English writing.
#Author’s response: Grammar and English writing of the text of the manuscript has been revised completely.
References
- Bharti, R., G. Dey, and M. Mandal, Cancer development, chemoresistance, epithelial to mesenchymal transition and stem cells: A snapshot of IL-6 mediated involvement. Cancer letters, 2016. 375(1): p. 51-61.
- Rich, J.N. and S. Bao, Chemotherapy and cancer stem cells. Cell stem cell, 2007. 1(4): p. 353-355.

Reviewer 2 Report
This study discusses the latest metabolic interventions by dual pathway inhibitors and the advantages and disadvantages of this therapeutic approach.
Comment: Please supply some information about the side effects of pathway inhibitors in the clinical studies
Author Response
#Reveiwer2
This study discusses the latest metabolic interventions by dual pathway inhibitors and the advantages and disadvantages of this therapeutic approach.
Comment: Please supply some information about the side effects of pathway inhibitors in the clinical studies
#Author’s response: We appreciate your time and effort. Clinical trials reported that common toxicities of administered PI3K/AKT/mTOR inhibitors were rash, gastrointestinal adverse events, fatigue, and asthenia. However, due to the impact of PI3K signaling on glucose metabolism, hyperglycemia may also occur [3]. However, other adverse events may also be reported following the administration of dual pathway inhibitors. These data have been added to Section 5.
References
- Markman, B., J. J Tao, and M. Scaltriti, PI3K pathway inhibitors: better not left alone. Current pharmaceutical design, 2013. 19(5): p. 895-906.

Reviewer 3 Report
This MS is very interesting since it presents the importance of developing novel effective and safe therapeutic interventions (e.g., aimed at the metabolism of tumor cells or immune cells, in the TME), especially for the most difficult-to-treat malignancies.
This review emphasizes the metabolic interventions by dual pathway inhibitors, including the advantages and disadvantages of this therapy.
The Authors may consider some suggestions for their minor revision.
Simple Summary and Abstract are identical – Perhaps, the ‘Simple Summary’ could include some helpful ‘take home points’ for the readers.
Table 1. should have a legend with an explanation of the most important abbreviations.
In the ‘Concluding remarks’, please, provide some examples:
‘The use of dual inhibitors of metabolic pathways can have a better effect in preventing the growth and progression of tumor cells due to the simultaneous inhibition of pathways such as the PI3K/AKT/mTOR pathway. However, in some cancers, [provide examples] the use of inhibitors of each pathway separately has had a better effect than dual inhibitors. Despite the various advantages, administrating dual inhibitors are associated with multiple challenges and limitations. For example, the mTOR pathway can sometimes trigger anti-tumor immune responses. In these cases, its inhibition will be related to the suppression of the immune system, and this issue can entirely depend on the tumor type [for example…], condition, and stage. Based on the available studies, it appears that combining dual inhibitors with other antitumor agents [for example…] can increase the effectiveness of the treatment.
Author Response
#Reveiwer3
This MS is very interesting since it presents the importance of developing novel effective and safe therapeutic interventions (e.g., aimed at the metabolism of tumor cells or immune cells, in the TME), especially for the most difficult-to-treat malignancies.
This review emphasizes the metabolic interventions by dual pathway inhibitors, including the advantages and disadvantages of this therapy.
The Authors may consider some suggestions for their minor revision.
- Simple Summary and Abstract are identical – Perhaps, the ‘Simple Summary’ could include some helpful ‘take home points’ for the readers.
#Author’s response: Thank you for your time and careful work. The simple Summary and Abstract have been revised based on your valuable comment.
- Table 1. should have a legend with an explanation of the most important abbreviations.
#Author’s response: A legend with the requested information has been added to Table 1.
- In the ‘Concluding remarks,’ please, provide some examples:
‘The use of dual inhibitors of metabolic pathways can have a better effect in preventing the growth and progression of tumor cells due to the simultaneous inhibition of pathways such as the PI3K/AKT/mTOR pathway. However, in some cancers, [provide examples] the use of inhibitors of each pathway separately has had a better effect than dual inhibitors. Despite the various advantages, administrating dual inhibitors are associated with multiple challenges and limitations. For example, the mTOR pathway can sometimes trigger anti-tumor immune responses. In these cases, its inhibition will be related to the suppression of the immune system, and this issue can entirely depend on the tumor type [for example…], condition, and stage. Based on the available studies, it appears that combining dual inhibitors with other antitumor agents [for example…] can increase the effectiveness of the treatment.
#Author’s response: Thank you again for your proposal. The requested data have been added to the Conclusion.
Pharmacological intervention in different metabolic pathways can lead to fundamental alterations in tumor cells' metabolism and pathological function, affecting immune responses in the TME. The use of dual inhibitors of metabolic pathways can have a better effect in preventing the growth and progression of tumor cells due to the simultaneous inhibition of pathways such as the PI3K/AKT/mTOR pathway. However, in some cancers, such as advanced pancreatic neuroendocrine tumors (pNET), the use of inhibitors of each pathway separately has had a better effect than dual inhibitors. Despite the various advantages, administrating dual inhibitors are associated with multiple challenges and limitations. For example, the mTOR pathway can sometimes trigger anti-tumor immune responses. In these cases, its inhibition may be associated with the suppression of the immune system, and this issue can entirely depend on the tumor type, signals, and stage. For instance, in melanoma, the PI3K/Akt, MyD88, and IKK pathways could be involved in IL-36β-mediated mTORC1 activation, promoting CD8+ T cell activation and inducing antitumor immune responses in vitro and in vivo [4]. Based on the available studies, it appears that combining dual inhibitors with other chemotherapeutic agents (paclitaxel and cisplatin) or other targeted therapies, such as trastuzumab or anti-immune checkpoint blockers (anti-PD-1 and anti-CTLA-4) can increase the effectiveness of the treatment [5-7]. However, common toxicities, especially gastrointestinal toxicities, and drug dose adjustment are also essential factors that should be considered in designing a pharmacologic protocol using monotherapy with dual inhibitors of metabolic pathways or combination therapies.
References
- Zhao, X., et al., IL-36β promotes CD8+ T cell activation and antitumor immune responses by activating mTORC1. Frontiers in Immunology, 2019. 10: p. 1803.
- Choi, H.J., et al., A novel PI3K/mTOR dual inhibitor, CMG002, overcomes the chemoresistance in ovarian cancer. Gynecologic Oncology, 2019. 153(1): p. 135-148.
- Rodon, J., et al., Phase 1/1b dose escalation and expansion study of BEZ235, a dual PI3K/mTOR inhibitor, in patients with advanced solid tumors including patients with advanced breast cancer. Cancer chemotherapy and pharmacology, 2018. 82: p. 285-298.
- Yan, C., et al., Inhibition of the PI3K/mTOR pathway in breast cancer to enhance response to immune checkpoint inhibitors in breast cancer. International Journal of Molecular Sciences, 2021. 22(10): p. 5207.
